# *Lactobacillus acidophilus* (LA) Fermenting *Astragalus* Polysaccharides (APS) Improves Calcium Absorption and Osteoporosis by Altering Gut Microbiota

**DOI:** 10.3390/foods12020275

**Published:** 2023-01-06

**Authors:** Junhua Zhou, Jing Cheng, Liu Liu, Jianming Luo, Xichun Peng

**Affiliations:** Department of Food Science and Engineering, Jinan University, Guangzhou 510632, China

**Keywords:** *Lactobacillus acidophilus*, *Astragalus* polysaccharides, fermentation, gut microbiota, calcium absorption, osteoporosis

## Abstract

*Lactobacillus acidophilus* (LA) and *Astragalus* polysaccharides (APS) have each been shown to have anti-osteoporotic activity, and the aim of this study was to further investigate whether the LA fermenting APS was more effective in improving calcium absorption and osteoporosis than the unfermented mixed solution (MS). We found that the fermentation solution (FS) intervention improved the calcium absorption, BMD, and bone microarchitecture in osteoporotic rats and resulted in better inhibition of osteoclast differentiation markers ACP-5 and pro-inflammatory cytokines TNF-α and IL-6 and promotion of osteoblast differentiation marker OCN. This better performance may be due to the improved restoration of the relative abundance of specific bacteria associated with improved calcium absorption and osteoporosis such as Lactobacillus, Allobaculum, and UCG-005. Several key metabolites, including indicaxanthin, chlorogenic acid, and 3-hydroxymelatonin, may also be the key to the better improvement. In conclusion, the LA fermenting APS can better improve calcium absorption and osteoporosis by increasing active metabolites and altering gut microbiota. This finding should become a solid foundation for the development of LA fermenting APS in functional foods.

## 1. Introduction

Osteoporosis is one of the most common non-communicable human diseases worldwide today and the most common disease of the adult skeleton. It is characterized by low bone mass and the destruction of bone tissue microstructure. Osteoporosis and the consequent fragility of fractures lead to a reduced quality of life and place a large and growing economic burden on individuals and societies around the world [1]. Inadequate calcium intake is a risk factor for osteoporosis. The daily calcium consumption in most low- and middle-income nations is far below guidelines, and poor intake is even reported in special age groups such as teenagers in high-income nations [2].

Commonly used drugs for osteoporosis treatment include bisphosphonates, estrogens, denosumab, and raloxifene, but all have certain side effects. For example, bisphosphonates can inhibit the resorption of bone by osteoclasts and simultaneously enhance osteoblast differentiation, thereby promoting bone formation. However, bisphosphonates have side effects including ulceration, osteonecrosis of the jaw, and musculoskeletal pain [3]. Calcium supplements are also commonly applied to treat osteoporosis. Inorganic calcium supplements have higher calcium content, but they may cause constipation and bloating and are poorly absorbed. Organic calcium supplements are better absorbed and have less impact on the intestinal tract, but they are more expensive than inorganic calcium supplements [4]. Therefore, there is a need to find a better and more economical way to improve calcium absorption for more effective prevention and relief of osteoporosis.

Osteoporosis can be effectively treated using non-digestible bioactive polysaccharides derived from natural plants with few side effects. In the stomach and small intestine, the host-derived enzymes were unable to decompose polysaccharides into an absorbable monosaccharide, but the majority of them passed into the large intestine and influenced its structure function through gut microbiota fermentation [5]. Gut microbiota has also been shown to be closely associated with osteoporosis and bone homeostasis metabolism. It can increase bone mass and improve osteoporosis by inhibiting osteoclast proliferation and differentiation, inducing apoptosis, reducing bone resorption, or promoting osteoblast proliferation and maturation [6]. Our previous study has shown that *Astragalus* polysaccharides (APS) can alleviate osteoporosis by improving structural changes in the gut microbiota [7].

Polysaccharides generally exert their effects in the intestine after being fermented by gut microbiota [8]. Some gut microbiota such as *Lactobacillus* can use polysaccharides to proliferate, and these probiotics have also been found to have an effect on improving osteoporosis [9]. For example, by promoting osteoblast differentiation and preventing osteoclast formation in OVX mice, supplementing with *Lactobacillus plantarum* GKM3 and *Lactobacillus paracasei* GKS6 can reduce bone loss [10]. In OVX mice, *Lactobacillus rhamnosus* can alter the balance of Treg-Th17 cells to prevent bone loss and maintain bone health [11]. By modulating the balance of Treg-Th17 cells, *Lactobacillus acidophilus* (LA) can prevent bone loss and promote osteoheterogeneity in osteoporotic rats [12].

Although LA and APS have each been shown to have anti-osteoporotic activity, no studies have been reported on whether they can act synergistically through fermentation to better alleviate osteoporosis. The purpose of the study was to investigate whether LA and APS could act synergistically with each other by improving the balance of gut microbiota and producing active metabolites to improve calcium absorption more effectively for more effective prevention and relief of osteoporosis.

## 2. Materials and Methods

### 2.1. Preparation of Crude APS

*Astragalus* was dehydrated and cut into slices. It was then extracted using 10 times (*v*/*w*) boiling distilled water twice for two hours. The liquid was collected by filtration through a gauze and concentrated in a vacuum rotary evaporator. Then, 4 times (*v*/*v*) absolute ethanol was added to the concentrate to precipitate the polysaccharides and placed in a 4 °C refrigerator for 12 h. The precipitated polysaccharides were collected by centrifugation (4000× *g*, 10 min) and washed three times with absolute ethanol. The precipitated polysaccharides were freeze-dried by vacuum, and the solution of crude APS was obtained after dissolving it in distilled water. The final content of APS was 68.4 g/L (phenol-vitriolic colorimetry method, standard curve: y = 9.73x − 0.001, R^2^ = 0.9996) [13].

### 2.2. Preparation of LA

The fermentation medium for LA was based on MRS medium with minor modifications. The specific recipe is as follows: glucose 10 g/L, peptone 10 g/L, beef extract 8 g/L, yeast extract 4 g/L, dipotassium hydrogen phosphate (K_2_HPO_4_) 2 g/L, diammonium hydrogen citrate (HOC(CO_2_H)(CH_2_CO_2_NH_4_)_2_) 2 g/L, sodium acetate (CH_3_COONa) 5 g/L, magnesium sulfate (MgSO_4_) 0.2 g/L, manganese sulfate (MnSO_4_) 0.04 g/L, and Tween 80 1 g/L. LA was inoculated at 8% (*v*/*v*), incubated at 37 °C with 180 r for 24 h, and stored at 4 °C.

### 2.3. Preparation of Fermentation Solution (FS)

The fermentation medium for FS was based on MRS medium with minor modifications. The specific recipe is as follows: APS 25 g/L, glucose 10 g/L, peptone 10 g/L, beef extract 8 g/L, yeast extract 4 g/L, dipotassium hydrogen phosphate (K_2_HPO_4_) 2 g/L, diammonium hydrogen citrate (HOC(CO_2_H)(CH_2_CO_2_NH_4_)_2_) 2 g/L, sodium acetate (CH_3_COONa) 5 g/L, magnesium sulfate (MgSO_4_) 0.2 g/L, manganese sulfate (MnSO_4_) 0.04 g/L, Tween 80 1 g/L, and calcium carbonate (CaCO_3_) 3 g/L. FS was inoculated with LA cultured in normal MRS medium at 8% (*v*/*v*) of the inoculum, incubated at 37 °C with 180 r for 24 h, and stored at 4 °C.

### 2.4. Analysis of Metabolomics in FS

In order to analyze whether LA fermenting APS can increase active ingredients to play a synergistic role in improving calcium absorption and osteoporosis, LC-MS-based untargeted metabolomics analysis was performed. FS was centrifuged at 12,000 rpm for 10 min after being inactivated at 85 °C. After being filtered with a 0.22 μM filter membrane, the supernatant was collected for analysis using liquid–liquid mass spectrometry. Using Acquity UPLC System I-Class PLUS (Waters, Milford, CT, USA) with a Waters Acquity UPLC Acquity UPLC HSS T3 column (1.8 μm, 2.1 mm × 100 mm) and a Xevo G2-XS QT high-resolution mass spectrometer (Waters, Milford, CT, USA) to perform untargeted metabolomics analysis. The raw LC-MS data were uploaded to the Majorbio cloud platform (https://cloud.majorbio.com, accessed on 15 September 2022) for data analysis after being pre-processed in the metabolomics processing program Progenesis QI (Waters Corporation, Milford, CT, USA). Significance of the differences between groups was analyzed using PLS-DA. Student’s *t*-test *p*-value and variable importance in projection (VIP) obtained from the OPLS-DA model were used to determine the selection of significantly different metabolites, with metabolites with *p* < 0.05 and VIP > 1 being significantly different metabolites.

### 2.5. Preparation of Mixed Solution (MS)

In order to analyze the differences before and after LA fermenting APS, MS was used as a control. APS, LA, and calcium carbonate were mixed well in proportion, the final concentrations of APS and calcium carbonate in MS were 25 and 3 g/L, and they were stored at 4 °C.

### 2.6. Animal Experiments

Twenty-four specific-pathogen-free (SPF) female Sprague Dawley (SD) rats (7~8 weeks, 180~200 g) were bought from Beijing HFK Bioscience Co. Ltd (Beijing, China). Rats were housed under controlled standard barrier conditions (temperature 23 ± 2 °C, humidity 55% ± 5, and 12 h light/dark cycle) at the Animal Center of Jinan University. After acclimatization for 1 week, rats were randomly divided into four groups (n = 6) and treated as follows: the Con group was intramuscularly injected with saline twice a week (8 weeks) and intragastrically administered sterile water daily. The other three groups were intramuscularly injected with dexamethasone (0.1 mg/100 g twice a week, 8 weeks) [14]. At the same time, sterile water was daily administered intragastrically to the Mod group, the Fer group was intragastrically administered 1 mL/100 g FS daily, and the Mix group was intragastrically administered 1 mL/100 g MS daily. The rats were fed in metabolic cages for 3 days before the end of the experiment, and the 3-day consumption of calcium including dietary intake and gavage volume were recorded. All feces were collected for the determination of apparent calcium absorption rate. The rats were then sedated with pentobarbital sodium before being executed via blood sampling through the abdominal aorta. Blood was drawn, and serum was centrifuged to determine ACP-5, OCN, TNF-α, and IL-6 levels. The right tibia was collected for histomorphological investigation and the left femur for BMD detection. For 16S rDNA sequencing and bioinformatics analysis, cecum contents were collected and kept at −80 °C.

All animal experiments were conducted with the agreement of Jinan University’s Ethics Committee (No. IACUC-20210526-07) and were in accordance with the established instructions.

### 2.7. Detection of Apparent Calcium Absorption

The rats were fed in metabolic cages 3 days before the end of the experiment, and the intake, gavage volumes and feces were recorded for 3 days. The feces were dried in an oven at 80 °C until they were of uniform weight and then cooled and finely ground in a desiccator. Calcium content was detected by ICP-MS. Apparent calcium absorption (%) = (calcium intake − fecal calcium)/calcium intake × 100%; calcium intake (g) = calcium content in feed (g/kg) × feed consumption (kg) + gavage dose (mL) × gavage calcium content (g/mL); fecal calcium (g/kg) = calcium content in feces (g/kg) × total mass of feces (kg) [15].

### 2.8. Detection of Bone Mineral Density (BMD)

Double-energy X-ray absorptiometry (Lunar iDXA, GE) was used to measure the BMD of left femurs, and the following exposure conditions were used: 0.0188 A, 100 kV, and 10.0 μGy.

### 2.9. Analysis of Tibia Paraffin Section, H&E Staining, and Bone Microarchitecture

The right tibia was removed and fixed for 3–5 days in 4% paraformaldehyde. Following that, decalcification, dehydration, removal, and paraffin immersion were carried out. Five μm thick slices were sectioned from paraffin-embedded samples and stained with hematoxylin and eosin (H&E). A Pannoramic MIDI digital scanner (3DHISTECH Ltd, Budapest, Hungary) was used to scan neutral balsam-mounted and coverslip-covered slides.

Image Pro Plus 6.0 software was used to calculate the tissue area (T.Ar), trabecular bone area (Tb.Ar), and trabecular bone perimeter (Tb.Pm). The bone volume/total volume (BV/TV) ratio, trabecular number (Tb.N), and trabecular separation (Tb.Sp) were computed as follows: BV/TV (%) = Tb.Ar/T.Ar × 100%, Tb.N (mm^−1^) = (1.199/2) × (Tb.Pm/T.Ar), Tb.Sp (μm) = (2000/1.199) × (T.Ar − Tb.Ar)/Tb.Pm [16].

### 2.10. Detection of ACP-5, OCN, TNF-α, and IL-6

At the time of execution, rat serum was collected by centrifugation (3000× *g*, 15 min) and kept at −80 °C before use. ACP-5, OCN, TNF-α, and IL-6 levels were measured using ELISA kits (Nanjing Jiancheng Bioengineering Institute Co., Ltd., Nanjing, China) as directed by the manufacturer.

### 2.11. Analysis of 16S rDNA Sequencing and Bioinformatics

After execution of the rats, the cecum contents were collected and stored at −80 °C until use. Microbial genomic DNA was isolated and purified from rat cecum contents, and PCR was used to amplify the 16s rDNA V3-V4 region. Illumina’s Miseq PE300/NovaSeq PE250 platform (Shanghai Meiji Biomedical Technology Co., Ltd., Shanghai, China) was used for sequencing after agarose gel electrophoresis, gel extraction, quantification, and library preparation. Using Usearch software (http://drive5.com/uparse/, version 7.1, accessed on 13 September 2022), the sequences were clustered with operational taxonomic units (OTUs) based on 97% similarity, chimeras were removed, and then the sequences taxonomically classified. The significance of the differences between groups was analyzed using PLS-DA. The classification of OTUs and genera was analyzed using a Venn diagram. Groups were compared in a ternary-component phase diagram, and LDA effect size (LEfSe) analysis was performed to identify bacterial markers (LDA score > 4) in each group. The relative abundance of gut microbiota between the groups was analyzed, predicting functional changes in gut microbiota of the Fer group osteoporotic rats with the KEGG database.

### 2.12. Statistics

The data were all presented as mean ± SD values. The online tool of Majorbio cloud platform (https://cloud.majorbio.com/page/tools/, accessed on 20 November 2022) was used to perform Spearman correlation analysis. GraphPad Prism 8 software (La Jolla, CA, USA) was used to carry out one-way analysis of variance or independent *t*-tests on the statistical differences. When *p* < 0.05, data from different groups of independent experiments were deemed statistically significant.

## 3. Results

### 3.1. FS More Significantly Improved Apparent Calcium Absorption in Osteoporotic Rats

Low levels of calcium absorption are an important cause of osteoporosis. The body weight and food intake of rats were normal (Appendix A). As shown in Figure 1, the apparent calcium absorption in the Mod group (30.581%) was significantly lower compared to the Con group (38.881%). Apparent calcium absorption in the Fer group (32.883%) was significantly improved, and between the Mix group (28.814%) and the Mod group, there was no significant difference (Figure 1). The results showed that FS more significantly improved the apparent calcium absorption in osteoporotic rats.

### 3.2. FS More Significantly Improved Rat Osteoporosis Induced by Dexamethasone

#### 3.2.1. FS More Significantly Improved BMD

BMD is the gold standard for the diagnosis of osteoporosis. As shown in Figure 2, BMD in the Mod group (0.180 g/cm^2^) was significantly lower compared to BMD in the Con group (0.209 g/cm^2^), indicating that dexamethasone intervention in osteoporotic rats was successful in modeling. BMD in the Fer group (0.206 g/cm^2^) was increased to comparable levels with the Con group and significantly higher than the Mod and the Mix group (0.186 g/cm^2^) (Figure 2). The results suggested that FS more significantly improved BMD in osteoporotic rats.

#### 3.2.2. FS More Significantly Improved the Restoration of Bone Microarchitecture

Impairment of bone microarchitecture is one of the major features of osteoporosis. As shown in Figure 3, compared with the Con group (Figure 3A), the Mod group (Figure 3B) had a smaller number of bone trabeculae with sparse distribution, and some of them were small and thin, locally connected into a network with a large number of gaps, indicating that dexamethasone intervention in osteoporotic rats was successful in modeling. Bone microarchitecture indexes such as the number, size, thickness, and spatial distribution of bone trabeculae were significantly restored in the Fer group (Figure 3C), while the restoration effect was average in the Mix group (Figure 3D). As shown in Table 1, Tb.Pm, BV/TV and Tb.N were significantly lower, Tb.Ar was lower but not significant, and Tb.Sp was significantly higher in the Mod group compared with the Con group, indicating a decrease in bone mass due to osteoporosis. In the Fer group, Tb.Pm, BV/TV, Tb.N, and Tb.Ar increased at higher levels, and Tb.Sp decreased at lower levels compared to the Mix group. The results demonstrated that the FS more significantly improved the restoration of bone microarchitecture.

### 3.3. FS More Significantly Decreased the Osteoclast Differentiation Biomarker ACP5 and Increased the Osteoblast Differentiation Marker OCN

ACP-5 and OCN are differentiation markers of osteoblasts and osteoclasts. Differentiation of osteoclasts for bone resorption and differentiation of osteoblasts for bone formation are critical in osteoporosis mitigation mechanisms. As shown in Figure 4, ACP-5 was significantly higher in the Mod group (5.891 ng/mL) than in the Con group (5.031 ng/mL) (Figure 4A), and OCN in the Mod group (4.914 ng/mL) showed a decreasing trend (not significant) compared with that in the Con group (5.815 ng/mL) (Figure 4B), indicating that dexamethasone significantly promoted osteoclast differentiation and had a tendency to inhibit osteoblast differentiation. ACP-5 (5.222 ng/mL) and OCN (6.018 ng/mL) in the Mix group were recovered to comparable levels with the Con group. ACP-5 (4.109 ng/mL) and OCN (8.570 ng/mL) in the Fer group were significantly lower and higher than the other three groups (Figure 4). The results showed that FS more significantly inhibited osteoclast differentiation and promoted osteoblast differentiation.

### 3.4. FS More Significantly Decreased the Levels of Pro-Inflammatory Cytokines TNF-α and IL-6

Pro-inflammatory cytokines such as TNF-α and IL-6 are among the high-risk factors in osteoporosis. As shown in Figure 5, TNF-α was significantly increased in the Mod group (170.807 ng/L) compared to the Con group (61.459 ng/L) and significantly reduced in the Fer group (112.939 ng/L) and did not recover significantly in the Mix group (152.235 ng/L) (Figure 5A). Similar to TNF-α, IL-6 was significantly increased in the Mod group (17.548 ng/L) compared to the Con group (8.466 ng/L) and significantly reduced in the Fer group (10.678 ng/L) and did not recover significantly in the Mix group (14.909 ng/L) (Figure 5B). The results presented that FS more significantly reduced the pro-inflammatory cytokines TNF-α and IL-6.

### 3.5. Changes in the Gut Microbiota Profile of Rats with Osteoporosis

#### 3.5.1. Composition Changes of Gut Microbiota in Osteoporotic Rats

The 16S rDNA sequencing of gut microbiota contents was used to explore the gut microbiota profiles of osteoporotic rats. With an OTU count ranging from 819 to 938, 1,211,432 clean tags (50,476 per sample) were generated. PLS-DA analysis (Figure 6A) showed that the sample points in the Con, Mod, Fer, and Mix groups were at completely non-overlapping locations, showing that their gut microbiota composition was different. We analyzed the data at the OTU and genus levels to gain a better understanding of the specific variances. All groups share 650 OTUs, and each group has its own set of OTUs (Figure 6B). The genus level analysis revealed that 11 appeared in the Mod group but not in the Con group (Figure 6C), five of them were specific to the Mod group; two appeared in both Mod and Fer groups; three appeared in both Mod and Mix groups; and one appeared in Mod, Fer, and Mix groups. All of the results provided above clearly demonstrated the substantial changes in gut microbiota composition in osteoporotic rats.

#### 3.5.2. Specific Bacteria Served as Bacterial Markers of FS to Improve Calcium Absorption and Osteoporosis

In order to identify changes in specific bacterial microbiota, linear discriminant analysis (LDA) effect size (LEfSe) was used to compare the microbiota composition of the Con, Mod, Fer, and Mix groups. The cladogram (Figure 7A) showed significant changes in the gut microbiota of the four groups under different taxa, and the LDA score histograms (Figure 7B) clearly showed the bacterial markers for each group.

Among the 31 bacterial markers shown in Figure 7B, o__Erysipelotrichales, f__Erysipelotrichaceae, g__*Allobaculum*, p__Actinobacteriota, c__Actinobacteria, f__Bifidobacteriaceae, o__Bifidobacteriales, g__*Bifidobacterium*, g__*Blautia*, and g__*Dubosiella* were specific to the Fer group. These bacteria could behave as bacterial markers in osteoporotic rats treated with FS, becoming intestinal metrics for calcium absorption and osteoporosis improvement.

### 3.6. Key Bacteria in FS-Improved Calcium Absorption and Osteoporosis

As shown in Figure 8, significant changes disparities between diverse groups at various taxonomic levels in the relative abundance of gut microbiota (phylum to genus, Figure 8). Among all the changes at genus level in Figure 8C, the relative abundance of *Lactobacillus*, *Allobaculum*, *UCG-005*, *Blautia*, *Christensenellaceae_R-7_group*, and *norank_f__Erysipelotrichaceae* was decreased, and the relative abundance of *unclassified_f__Lachnospiraceae*, *Lachnospiraceae_NK4A136_group*, *norank_f__Lachnospiraceae*, *Lachnoclostridium*, *Ruminococcus*, *unclassified_f__Oscillospiraceae*, *norank_f__Muribaculaceae*, *norank_f__Oscillospiraceae*, and *Roseburia* was increased in the Mon group when compared with the Con group (Table 2). These changes were all recovered in the Fer group, and the degree of recovery was greater than that in the Mix group; these 15 genera were regarded to be the key bacteria.

### 3.7. Functional Changes of Gut Microbiota Related to Calcium Absorption in Rats with Osteoporosis Treated by FS

Through specific metabolic pathways, various gut microbial compositions might produce distinctive metabolites and engage in interactions with hosts with various physiological functions, so we predicted the calcium-absorption-related functions of FS to improve gut microbiota through composition changes. PICRUSt was applied to predict microbial metabolic function and to analyze functional differences between groups according to the KEGG database. As shown in Table 3, KEGG analysis revealed that among the top 30 pathways in relative abundance, eight pathways were restored in the Fer group, namely carbohydrate metabolism, amino acid metabolism, metabolism of cofactors and vitamins, cell motility, cell growth and death, endocrine system, aging and immune system, and the degree of recovery, which was greater than that in the Mix group. The results demonstrated that FS may improve calcium absorption function in osteoporotic rats by interfering with the composition of the gut microbiota to restore the function of relevant pathways.

### 3.8. Key Differential Metabolites of FS to Improve Calcium Absorption and Osteoporosis

To analyze the active ingredients in FS, untargeted metabolomics analysis using LC-MS was conducted between FS and MS. PLS-DA analysis (Figure 9A) revealed that sample points in the Fer and Mix groups were at completely non-overlapping positions, indicating that the metabolite composition was significantly different. Further analysis of the differential metabolites between the two groups was screened by variable importance in projection (VIP) > 1 and *p* < 0.05. Among the 5166 total metabolites, 368 were significantly different, 251 had significantly higher relative metabolite content, and 117 had significantly lower relative metabolite content (Figure 9B). Using HMDB compound classification analysis (Figure 9C), according to the Superclass hierarchy, these differential metabolites were divided into 10 classes, of which 78 were annotated as organic acids and derivatives, 57 as lipids and lipid-like molecules, 52 as organoheterocyclic compounds, and others as organic oxygen compounds and nucleosides, nucleotides, and analogues. Three upregulated differential metabolites, indicaxanthin, chlorogenic acid, and 3-hydroxymelatonin, were detailed according to the fold change (FC) of differential metabolites between groups (Table 4), inferring that these three are key metabolites in FS to improve calcium absorption and osteoporosis.

## 4. Discussion

LA and APS have been shown to exert anti-osteoporosis activity by improving gut microbiota composition. In this study, we demonstrated that LA fermenting APS can increase calcium absorption and had anti-osteoporotic activities such as BMD restoration and bone microarchitecture repair in a dexamethasone-induced osteoporosis rat model.

Calcium malabsorption is a major feature of osteoporosis. A previous study has shown that restoring intestinal calcium absorption can improve osteoporosis and bone loss [17]. In this study, the apparent calcium absorption in osteoporotic rats treated with FS was significantly increased. This demonstrated that the FS significantly restored calcium absorption in osteoporotic rats by combining LA and APS after fermenting and contributed to the restoration of BMD and bone microarchitecture. Bone is a dynamic tissue that continuously undergoes modeling and remodeling processes to maintain strength and robustness. Bone remodeling is regulated by both osteoblasts and osteoclasts, and an imbalance between the two can lead to osteoporosis [18]. Dexamethasone has the function of inhibiting osteoblast differentiation and promoting osteoclast differentiation [19,20] and is commonly used to cause osteoporosis in animal experiments. [21]. In this study, the osteoclast differentiation marker ACP-5 [22] was significantly increased, and the osteoblast differentiation marker OCN [23] tended to decrease via dexamethasone. ACP-5 decreased more significantly, and OCN increased more significantly after FS treatment. This suggests that FS was more effective in inhibiting osteoclast differentiation and promoting osteoblast differentiation and may contribute to restoration of osteoporosis. There is a strong link between inflammation and osteoporosis. Pro-inflammatory cytokines can disrupt the balance between osteoclasts and osteoblasts, causing bone loss and osteoporosis [24]. TNF-α can lead to impaired bone formation by inhibiting osteoblast differentiation; this plays a central role in the pathogenesis of postmenopausal osteoporosis [25]. IL-6 can cause osteoporosis by stimulating osteoclastogenesis to promote bone resorption [26]. In this study, FS reduced the levels of pro-inflammatory cytokines TNF-α and IL-6 more significantly. All these results demonstrated that LA fermenting APS can better restore calcium absorption and may achieve better improvement of osteoporosis by inhibiting osteoclast differentiation, promoting osteoblast differentiation, and inhibiting inflammation.

Recent studies have shown that gut microbiota and osteoporosis are closely linked and that restoring gut microbiota balance can improve osteoporosis [6,27]. At the same time, the intestine is the only channel through which calcium enters the human and mammalian body. Dysbiosis of the ecological composition of gut microbiota can affect calcium absorption by affecting intestinal permeability and metabolic disorders [28]. Both LA and APS have been shown to function as interactions with the gut microbiota. We therefore proposed a key role for bacteria in the gut microbiota to improve calcium absorption and alleviate osteoporosis after LA fermenting APS. In this study, we inferred that 15 genera, namely *Lactobacillus*, *Allobaculum*, *UCG-005*, *Blautia*, *Christensenellaceae_R-7_group*, *norank_f__Erysipelotrichaceae*, *unclassified_f__Lachnospiraceae*, *Lachnospiraceae_NK4A136_group*, *norank_f__Lachnospiraceae*, *Lachnoclostridium*, *Ruminococcus*, *unclassified_f__Oscillospiraceae*, *norank_f__Muribaculaceae*, *norank_f__Oscillospiraceae*, and *Roseburia*, could be the key bacteria in gut microbiota to improve calcium absorption and alleviate osteoporosis.

Among the bacteria with upregulated relative abundance after FS intervention, *Lactobacillus* was widely reported to enhance calcium absorption in the intestine. For example, calcium transport and uptake by intestinal cells in vitro were improved by *Lactobacillus* from Mongolia [29], and probiotic yogurt containing *Lactobacillus casei*, *Lactobacillus reuteri*, and *Lactobacillus gasseri* strains increased apparent calcium absorption in rats [30]. In addition, it was found that increased abundance of *Allobaculum* and *Blautia* could promote the production of SCFAs to lower intestinal pH and thus promote calcium absorption [31]. At the same time, *Allobaculum* was shown to be negatively correlated with the pro-inflammatory cytokines TNF-α and IL-6 [32], while *Blautia* can attenuate inflammatory and metabolic diseases [33], demonstrating that *Allobaculum* and *Blautia* have anti-inflammatory effects. Similarly, *UCG-005* can affect a variety of metabolic functions through the production of butyric acid [34], which can expand the intestinal absorption area and promote calcium absorption [35]. *Christensenellaceae_R-7_group* can promote the conversion of carbohydrates to acetate [36], which is used to enhance intestinal calcium absorption and thus reduce ovariectomy-induced bone conversion [37]. Among the bacteria with downregulated relative abundance after FS intervention, *Lachnospiraceae_NK4A136_group* was shown to be negatively associated with intestinal permeability, thereby protecting the intestinal barrier [38]. In addition, *Lachnoclostridium* was found to possess higher abundance in patients with osteopenia and osteoporosis [39]. The relative abundance of *Ruminococcus* was positively correlated with osteoclastic indices, and *Roseburia* showed an increasing trend after osteoporosis-modeling OVX resection [40]. In this study, LA fermenting APS restored or reversed the relative abundance of these bacteria better than the unfermented mixture samples. All these results demonstrated that LA fermenting APS could better improve osteoporosis by regulating key bacteria to exert effects such as promoting calcium absorption.

Correlation analysis was performed to evaluate the roles of each of the potentially key bacteria. As shown in Figure 10, the relative abundances of *Lactobacillus*, *Blautia*, *Christensenellaceae_R-7_group*, and *norank_f__Erysipelotrichaceae* were positively correlated with apparent calcium absorption, BMD, Tb.Ar, Tb.Pm, BV/TV, and Tb.N, while the relative abundances of *Ruminococcus*, *unclassified_ f__Oscillospiraceae*, *norank_f__Muribaculaceae*, and *norank_f__Oscillospiraceae* were negatively correlated with apparent calcium absorption, BMD, and Tb.Sp. Among them, *Blautia* showed a strong positive correlation (|R| ≥ 0.5) with both calcium absorption and BMD, and *unclassified_f__Oscillospiraceae* and *norank_f__Oscillospiraceae* showed a strong negative correlation (|R| ≥ 0.5), demonstrating that changes in the relative abundance of these bacteria may play a key role in improving calcium absorption and BMD and repairing bone microstructures. In addition, among the relevant indicators of bone metabolism and pro-inflammatory cytokines, *unclassified_f__Oscillospiraceae* was strongly positively correlated with ACP-5 (|R| ≥ 0.5), *Lachnoclostridium* was strongly negatively correlated with OCN (|R| ≥ 0.5), and *unclassified_f__Lachnospiraceae*, *norank_f__Lachnospiraceae*, and *Roseburia* were strongly positively with ACP-5 and strongly negatively correlated with OCN (|R| ≥ 0.5) at the same time. *Lactobacillus* and *Blautia* showed a strong negative correlation with TNF-α (|R| ≥ 0.5), *norank_f__Oscillospiraceae* showed a strong positive correlation with IL-6 (|R| ≥ 0.5), and *unclassified_f__Oscillospiraceae* showed a strong positive correlation with both TNF-α and IL-6 (|R| ≥ 0.5). These results indicate that changes in the relative abundance of these bacteria play a key role in better improvement of osteoporosis. The correlation analysis further revealed the key role of potential key bacterial changes induced by LA fermenting APS in promoting calcium absorption for better osteoporosis, and further studies are worthwhile to determine how these bacteria act on the host and are involved in improving osteoporosis.

Based on gut microbiota composition changes and the various bacteria activities, we inferred that LA fermenting APS affected the functional changes of the gut microbiota in osteoporotic rats. KEGG analysis showed that many pathways were better altered or restored by FS, and these pathways may be involved in facilitating the restoration of calcium-absorption-related functions to improve osteoporosis. For example, the carbohydrate metabolic pathway regulates the metabolism of intestinal carbohydrates such as SCFAs [41]. SCFAs have been shown to directly affect calcium absorption and increase its transport by modulating signaling pathways [42]. The amino acid metabolic pathway regulates the metabolism of intestinal amino acids. Gut microbiota can metabolize some amino acids into a series of active metabolites to regulate the viability of the intestinal epithelium and the integrity of the barrier, thereby improving the efficiency of calcium absorption [43]. Metabolism of cofactors and the vitamin pathway regulates the metabolism of intestinal vitamin metabolism, especially vitamin D, playing a key role in stimulating intestinal absorption of calcium and promoting bone health [44]. Further validation is needed as to whether LA fermenting APS actually alters or restores these pathways to promote calcium absorption function to improve osteoporosis.

Recent studies have shown that *Lactobacillus* fermenting polysaccharides can be more beneficial to improving host health by increasing the bioactive substrates [45,46]. In this study, we screened three key differential metabolites increased by FS, namely indicaxanthin, chlorogenic acid, and 3-hydroxymelatonin, playing a key role in improving calcium absorption and osteoporosis. Indicaxanthin can help control or mitigate the inflammatory response in chronic inflammatory bowel disease and help maintain the intestinal barrier and permeability [47], and restoring intestinal barrier and permeability can effectively improve calcium absorption [48]. Chlorogenic acid can improve the BMD and trabecular micro-architecture for OVX-induced osteoporosis [49]. 3-Hydroxymelatonin is a metabolite of melatonin. It can act as an inhibitor of osteoclast activity in bone depending on its free radical scavenging properties and antioxidant properties [50]. These results reveal that LA fermenting APS can act synergistically by increasing beneficial metabolites to better improve calcium absorption and osteoporosis.

## 5. Conclusions

In summary, our study identified that LA fermenting APS improved calcium absorption and osteoporosis more effectively than the unfermented mixed solution. The fermentation solution intervention more significantly restored the relative abundance of specific bacteria associated with improved calcium absorption and osteoporosis in gut microbiota of osteoporotic rats including *Lactobacillus*, *Allobaculum*, *UCG-005*, *Blautia*, *Christensenellaceae_R-7_group*, *norank_f__Erysipelotrichaceae*, *unclassified_f__Lachnospiraceae*, *Lachnospiraceae_NK4A136_group*, *norank_f__Lachnospiraceae*, *Lachnoclostridium*, *Ruminococcus*, *unclassified_f__Oscillospiraceae*, *norank_f__Muribaculaceae*, *norank_f__Oscillospiraceae*, and *Roseburia.* These bacteria were also closely associated with the restoration of osteoblast differentiation marker OCN and the inhibition of the osteoclast differentiation marker ACP-5 as well as pro-inflammatory cytokines TNF-α and IL-6. Furthermore, increased active metabolites from fermentation, including indicaxanthin, chlorogenic acid, and 3-hydroxymelatonin, are also key to improvement. Our study provided a potential target for LA fermenting APS to regulate specific gut microbiota and metabolites to improve calcium absorption and osteoporosis and a strong base for the creation and use of functional foods to improve calcium absorption and osteoporosis.

## Figures and Tables

**Figure 1 foods-12-00275-f001:**
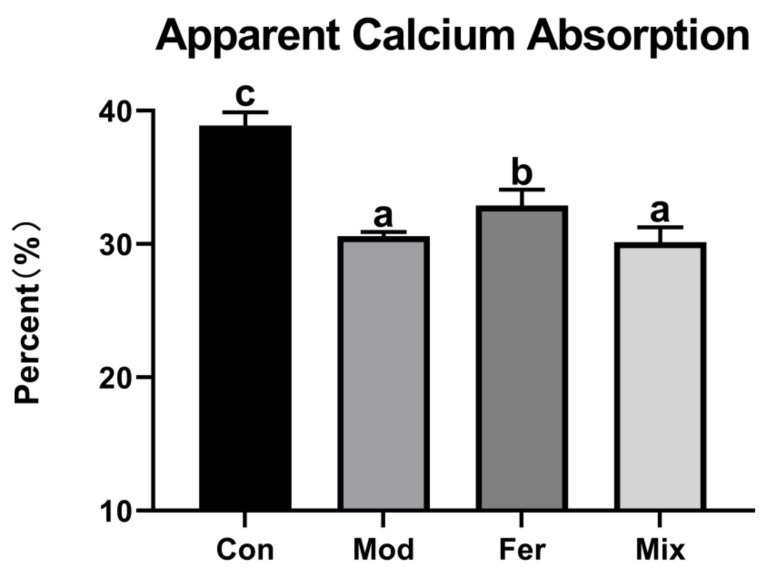
FS more significantly restored apparent calcium absorption in osteoporotic rats. Rats in each group were fed in metabolic cages 3 days before the end of the experiment, and the intake, gavage volumes, and feces were recorded for 3 days. Calcium content was detected by ICP-MS. Apparent calcium absorption (%) = (calcium intake − fecal calcium)/calcium intake × 100%; calcium intake (g) = calcium content in feed (g/kg) × feed consumption (kg) + gavage dose (mL) × gavage calcium content (g/mL); fecal calcium (g/kg) = calcium content in feces (g/kg) × total mass of feces (kg). Values are expressed as mean ± SD (n = 6). With *p* < 0.05, values with various superscript letters differ significantly.

**Figure 2 foods-12-00275-f002:**
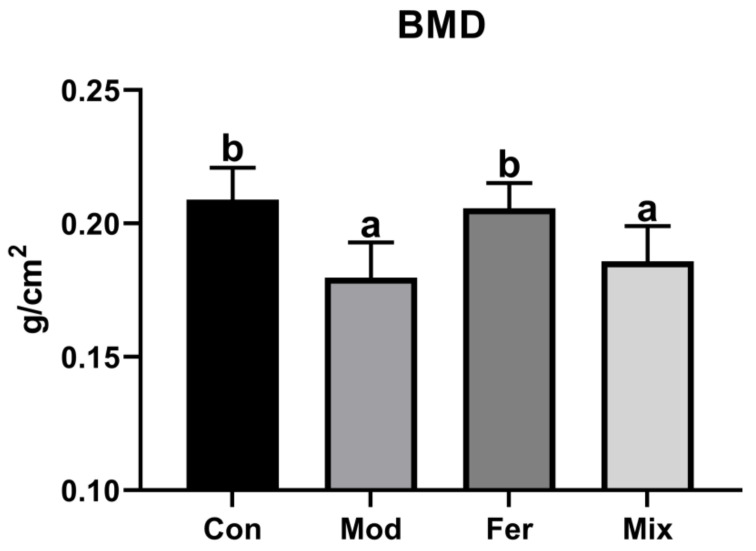
FS more significantly restored BMD in osteoporotic rats. Three groups of female rats were injected intramuscularly with dexamethasone to induce osteoporosis, and the rats in the Con group were injected with an equal volume of saline. Meanwhile, the Fer group was intragastrically administered 1 mL/100 g of FS, the Mix group was intragastrically administered 1 mL/100 g of MS, and the Con and Mod groups were intragastrically administered an equal volume of sterile water. After execution, double-energy X-ray absorptiometry (Lunar iDXA, GE) was used to analyze the BMD of the left femur. The values are given as mean ± SD (n = 6). With *p* < 0.05, values with various superscript letters differ significantly.

**Figure 3 foods-12-00275-f003:**
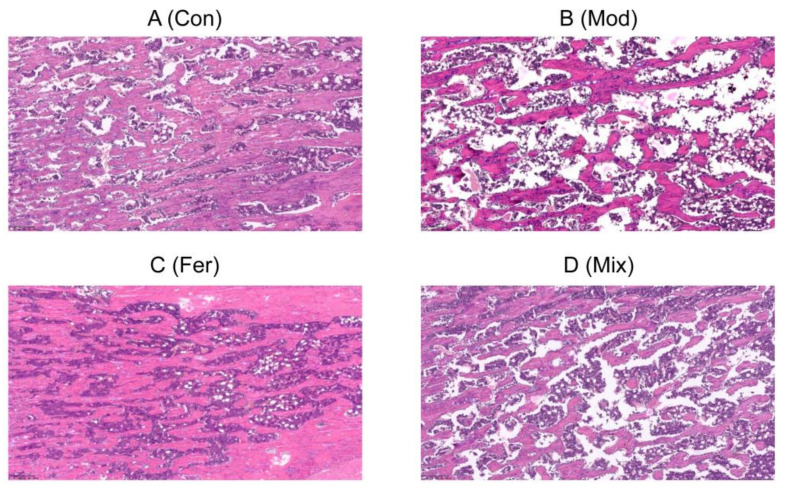
FS repaired the impairment of bone microarchitecture in osteoporotic rats. Representative pathological sections of the Con group (**A**), Mod group (**B**), Fer group (**C**) and Mix group (**D**). The Con, Mod, Fer, and Mix groups’ right tibias were sampled and prepared for paraffin sectioning, H&E staining, and bone microarchitecture.

**Figure 4 foods-12-00275-f004:**
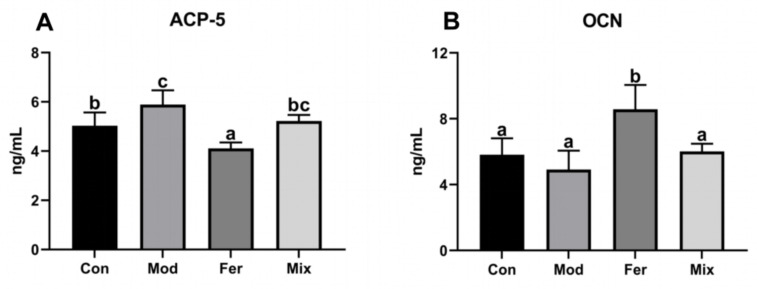
Serum levels of ACP-5 (**A**) and OCN (**B**). Serum of rats was collected at the time of execution, and the levels of ACP-5 and OCN were measured using ELISA kits. Values are expressed as mean ± SD (n = 6). With *p* < 0.05, values with various superscript letters differ significantly.

**Figure 5 foods-12-00275-f005:**
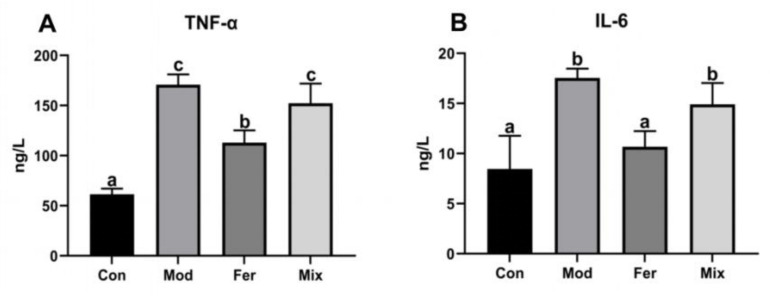
Serum levels of TNF-α (**A**) and IL-6 (**B**). Serum of rats was collected at the time of execution, and the levels of TNF-α and IL-6 were measured using ELISA kits. Values are expressed as mean ± SD (n = 6). With *p* < 0.05, values with various superscript letters differ significantly.

**Figure 6 foods-12-00275-f006:**
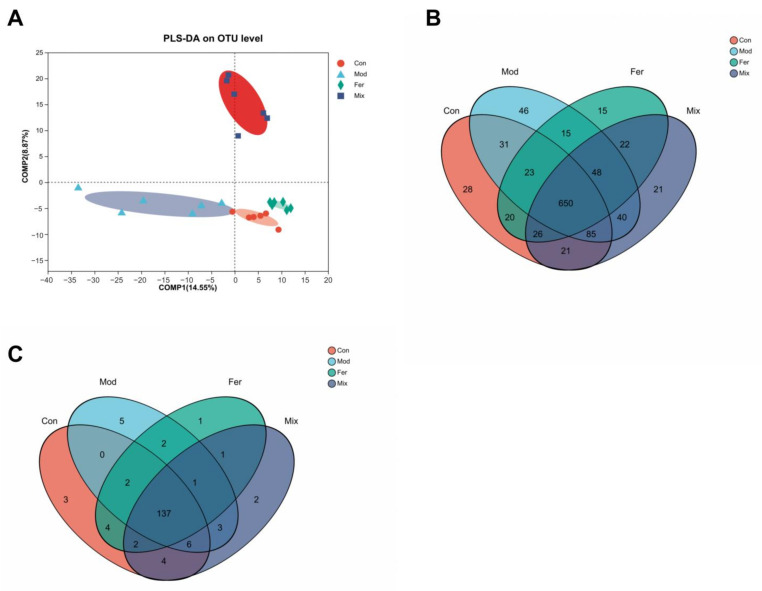
Gut microbiota diversity in different groups. The contents of rats’ ceca were collected, sequenced, and evaluated for community structure. Differences in gut microbiota diversity across groups were determined by using PLS−DA analysis (**A**). The Venn diagram depicts the number of common and individual OTUs (**B**) and genera (**C**).

**Figure 7 foods-12-00275-f007:**
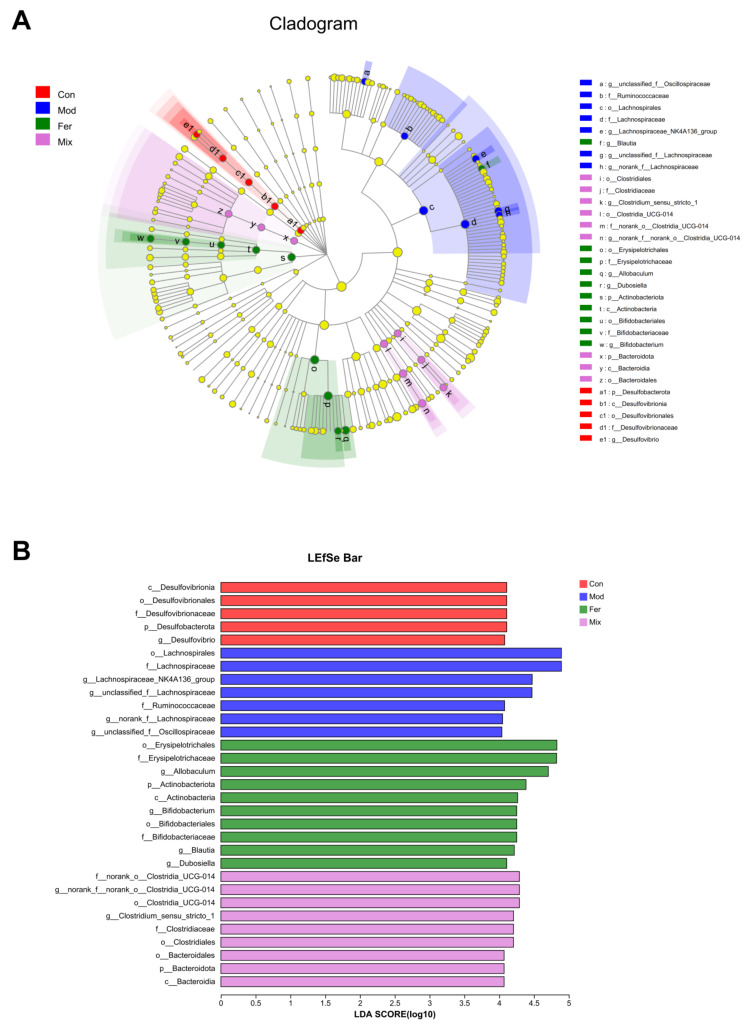
Changes in gut microbiota at different levels and interactions between them. LEfSe analysis was performed to identify the taxa that changed significantly. A cladogram (**A**) was used to analyze the changes in the gut microbiota occurring in each group across taxa, and LDA score histograms (**B**) were used to analyze the bacterial markers in each group.

**Figure 8 foods-12-00275-f008:**
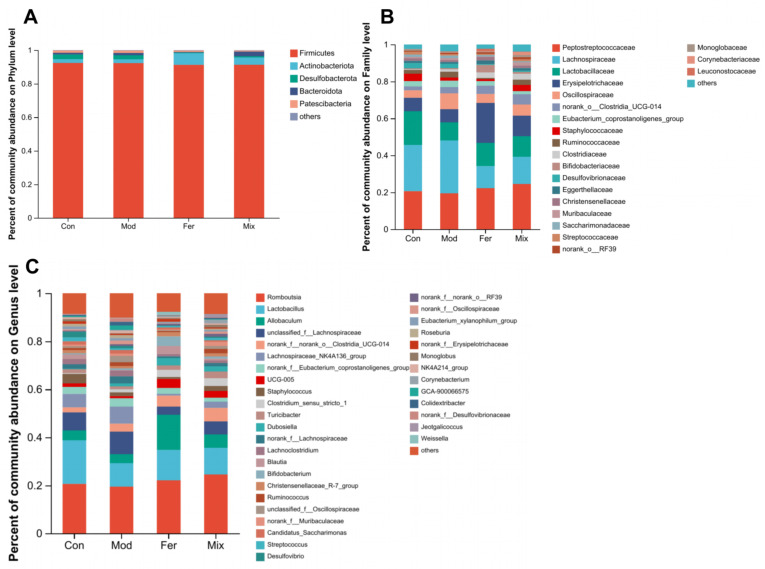
Composition at the level of phylum (**A**), family (**B**), and genus (**C**) of gut microbiota. Relative abundance of different group of gut microbiota was analyzed at the genus level to identify potential key bacteria.

**Figure 9 foods-12-00275-f009:**
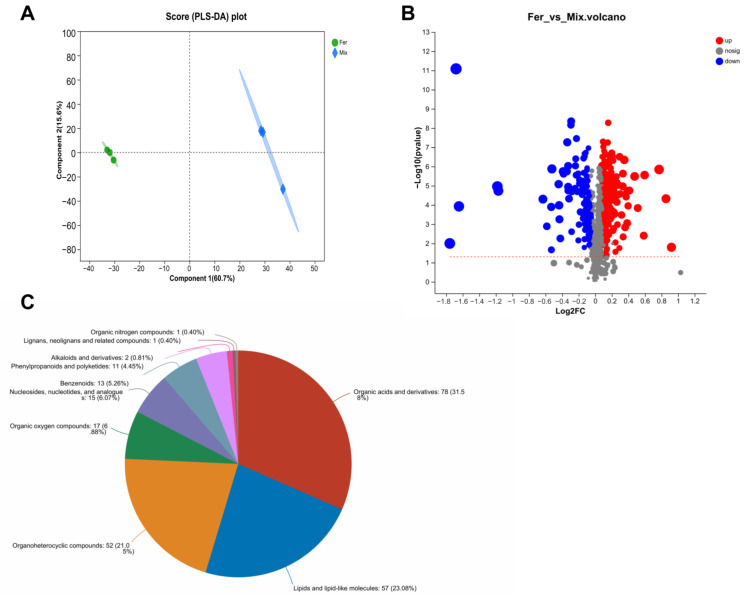
Analysis of differential metabolites. PLS−DA analysis (**A**) was performed to show the variation in metabolites diversity between the Fer and Mix groups. A volcano plot (**B**) was used to analyze the differential metabolites in the Fer and Mix groups. Gray dots denote non−significant differential metabolites, blue dots denote significantly downregulated metabolites (*p* < 0.05), and red dots denote significantly upregulated metabolites (*p* < 0.05). HMDB compound classification analysis (**C**) was used to classify and annotate differential metabolites.

**Figure 10 foods-12-00275-f010:**
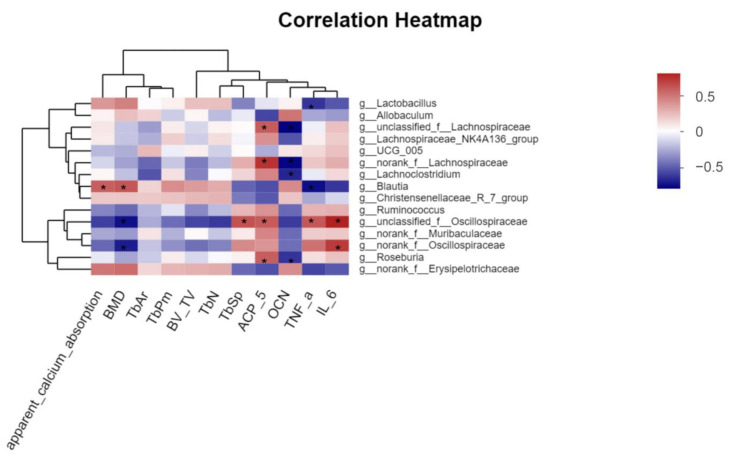
Correlation analysis between other parameters and the relative abundance of inferred key bacteria. The Majorbio cloud platform’s web tool (https://cloud.majorbio.com/page/tools/, accessed on 20 November 2022) was utilized to perform correlation analysis, and R values are indicated by colors with different depths (* |R| ≥ 0.5).

**Table 1 foods-12-00275-t001:** Right tibial bone histomorphometry.

	Con	Mod	Fer	Mix
Tb.Ar (mm^2^)	8.373 ± 2.482 ^a^	6.422 ± 1.841 ^a^	8.239 ± 1.698 ^a^	7.705 ± 1.577 ^a^
Tb.Pm (mm)	283.528 ± 62.196 ^b^	169.300 ± 24.429 ^a^	233.897 ± 77.859 ^ab^	171.904 ± 21.450 ^a^
BV/TV (%)	65.613 ± 22.370 ^b^	32.837 ± 11.319 ^a^	53.187 ± 21.070 ^ab^	39.872 ± 8.866 ^ab^
Tb.N (mm^−1^)	13.296 ± 3.709 ^b^	5.288 ± 1.442 ^a^	8.646 ± 2.573 ^a^	5.369 ± 1.005 ^a^
Tb.Sp (μm)	28.644 ± 19.799 ^b^	136.691 ± 38.530 ^a^	59.987 ± 32.019 ^b^	119.035 ± 42.584 ^a^

Image Pro Plus 6.0 software was used to obtain T.Ar, Tb.Ar, and Tb.Pm. BV/TV ratio, Tb.N, and Tb.Sp were calculated. Values are expressed as mean ± SD (n = 6). With *p* < 0.05, values with various superscript letters differ significantly.

**Table 2 foods-12-00275-t002:** Gut microbiota relative abundance at the genus level.

Genus	Relative Abundance (%)			
	Con	Mod	Fer	Mix
*Lactobacillus*	18.2040	9.7956	12.6682	11.1472
*Allobaculum*	4.0409	3.7342	14.6471	5.5833
*unclassified_f__Lachnospiraceae*	7.5366	9.3171	3.4125	5.4050
*Lachnospiraceae_NK4A136_group*	5.5491	7.0952	0.7910	2.5797
*UCG-005*	1.4815	0.8556	3.7246	2.8082
*norank_f__Lachnospiraceae*	1.9242	2.9739	0.6871	1.3591
*Lachnoclostridium*	2.0744	2.3404	0.9932	1.0434
*Blautia*	1.9770	0.2813	3.3311	0.7532
*Christensenellaceae_R-7_group*	1.2659	0.9645	1.7921	1.3322
*Ruminococcus*	0.9681	1.6450	0.4389	1.8254
*unclassified_f__Oscillospiraceae*	0.6484	2.5963	0.3425	1.2051
*norank_f__Muribaculaceae*	0.7546	1.0063	0.2851	1.7442
*norank_f__Oscillospiraceae*	0.5453	1.5403	0.1912	0.9099
*Roseburia*	1.1361	1.3167	0.1218	0.5946
*norank_f__Erysipelotrichaceae*	1.0560	0.4637	1.1465	0.4231

**Table 3 foods-12-00275-t003:** Functional pathways of gut microbiota were altered by FS.

Pathways	Relative Abundance (%)			
	Con	Mod	Fer	Mix
KEGG: Carbohydrate metabolism	10.4620	10.1761	10.5526	10.3166
KEGG: Amino acid metabolism	6.5946	6.8001	6.7853	6.8195
KEGG: Metabolism of cofactors and vitamins	3.7919	3.8326	3.7920	3.8840
KEGG: Cell motility	1.0894	1.5041	0.8104	1.0483
KEGG: Cell growth and death	0.7518	0.7693	0.7483	0.7595
KEGG: Endocrine system	0.6319	0.6315	0.6638	0.6498
KEGG: Aging	0.2654	0.2752	0.2680	0.2878
KEGG: Immune system	0.2660	0.2925	0.2607	0.2754

**Table 4 foods-12-00275-t004:** The FC of key upregulated differential metabolites between the Fer and Mix groups.

Differential Metabolite	FC(Fer/Mix)	*p* Value	Regulated
Indicaxanthin	1.2746	0.000000453	up
Chlorogenic Acid	1.2231	0.01733	up
3-Hydroxymelatonin	1.1687	0.0002299	up

## Data Availability

Data is contained within the article or Appendix A.

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
