# Peer review of "Lactobacillus acidophilus (LA) Fermenting Astragalus Polysaccharides (APS) Improves Calcium Absorption and Osteoporosis by Altering Gut Microbiota"

_foods, 2023, doi:10.3390/foods12020275_

Round 1
Reviewer 1 Report
Dear Editor, This manuscript ‘Lactobacillus acidophilus (LA) fermenting Astragalus polysaccharides (APS) improves calcium absorption and osteoporosis by altering gut microbiota’, by Junhua Zhou et al. is interesting and well-presented research that might be useful for future readers, also this manuscript is an adequate subject for your Journal of foods, MDPI. Where the overall subject is meaningful and worthy of study. This MS requires further refinement in the content of the information presented. The scope of this manuscript is clear but needs to be more supported by scientific data. However, the manuscript can be accepted With some global improvement changes for publication in its current version to fulfill the quality requirements.
- Abstract: informative but too long. Please shorten this part. As well as, The Abstract is not well presented, please rewrite it. As a proposition for Author, The author should begin the Abstract with the purpose of the research, then present their results explaining in which consists their originality, and finally explain the importance of these results obtained from those extracts of selected medicinal plant species.
- The introduction: should be revised by English Editor.
- The discussion of the results is presented in general, I suggest rewriting it to present the result of the work by interpreting the results at a higher level of abstraction from the discussion and linking these results to the objective given in the introduction and perhaps giving a future view of this study, in order to make the article more interesting for the readers.
- The discussion of the results is presented in general, I suggest rewriting it to present the result of the work by interpreting the results at a higher level of abstraction from the discussion and linking these results to the objective given in the introduction and perhaps giving a future view of this study, in order to make the article more interesting for the readers.
- The fig.86,7,8,9 and10 are not clear.
- Figure 1 and 2 are best placed in the supplementary file.
- Revise the conclusion section please and make it more appropriate, in addition, revise the manuscript for all typological and grammatical errors.
- Include list of abbreviations
- The format of references shall meet the requirements of FOODS journals. The journal name should be abbreviated.
Sincerely
Author Response
Dear Reviewer:
Thank you for your letter and for giving us an opportunity to revise the manuscript. “Lactobacillus acidophilus (LA) fermenting Astragalus polysaccharides (APS) improves calcium absorption and osteoporosis by altering gut microbiota” (Manuscript Number: foods-2123939). We appreciate the positive and constructive comments from the reviewers, which has significantly raised the quality of the manuscript and enable us to improve the manuscript. We have studied comments carefully and have made correction. Revised portion are marked in red in the paper.The main corrections in the paper and the responds to the reviewer’s comments are as follows:
- Abstract: informative but too long. Please shorten this part. As well as, The Abstract is not well presented, please rewrite it. As a proposition for Author, The author should begin the Abstract with the purpose of the research, then present their results explaining in which consists their originality, and finally explain the importance of these results obtained from those extracts of selected medicinal plant species.
Response: Thanks for the kindly suggestion. Abstract has been shortened and rewritten.
- The introduction: should be revised by English Editor.
Response: Thanks for the kindly suggestion. The introduction has been checked for grammatical errors and corrections.
- The discussion of the results is presented in general, I suggest rewriting it to present the result of the work by interpreting the results at a higher level of abstraction from the discussion and linking these results to the objective given in the introduction and perhaps giving a future view of this study, in order to make the article more interesting for the readers.
Response: Thanks for the kindly suggestion. The discussion has been rewritten more appropriately.
- The fig.86,7,8,9 and10 are not clear.
Response: Thanks. The Fig. 6, 7, 8, 9 and 10 have been reinserted and enlarged.
- Figure 1 and 2 are best placed in the supplementary file.
Response: Thanks for the kindly suggestion, but apparent calcium absorption and BMD are the two most direct indicators of the effect of improving calcium absorption and osteoporosis, so we think Fig. 1 and Fig. 2 should be left in the article.
- Revise the conclusion section please and make it more appropriate, in addition, revise the manuscript for all typological and grammatical errors.
Response: Thank you for the nice advice. The conclusion has been revised and the grammatical errors in the manuscript have been corrected.
- Include list of abbreviations
Response: Thanks for the nice advice. The abbreviations have been added to the end of the manuscript.
- The format of references shall meet the requirements of FOODS journals. The journal name should be abbreviated.
Response: Thanks. The format of references has been modified according to the requirements of FOODS journals.
Thank you and best regards.
Yours sincerely,
Xichun Peng, Ph.D
Jianming Luo, Ph.D
Email: [email protected]; [email protected]

Reviewer 2 Report
Abstract: FS intervention appears the first time explain it more clearly.
Material method: 2.3. What author means by “FS was inoculated”? Can the author write it more clear?
I think the author should perform an experiment with a similar animal model to check the individual & conbination effect of Indicaxanthin, Chlorogenic Acid and 3-Hydroxymela-tonin on the osteoporosis.
I would like to add primary data from animal experiments, may in supplementary figures, such as water or food intake, body weight etc.
Author Response
Dear Reviewer:
Thank you for your letter and for giving us an opportunity to revise the manuscript. “Lactobacillus acidophilus (LA) fermenting Astragalus polysaccharides (APS) improves calcium absorption and osteoporosis by altering gut microbiota” (Manuscript Number: foods-2123939). We appreciate the positive and constructive comments from the reviewers, which has significantly raised the quality of the manuscript and enable us to improve the manuscript. We have studied comments carefully and have made correction. Revised portion are marked in red in the paper.The main corrections in the paper and the responds to the reviewer’s comments are as follows:
- Abstract: FS intervention appears the first time explain it more clearly.
Response: Thanks. Fermentation solution (FS) has been explained clearly in the abstract.
- Material method: 2.3. What author means by “FS was inoculated”? Can the author write it more clear?
Response: Thanks for the kindly suggestion. “FS was inoculated” means FS was inoculated with LA cultured in normal MRS medium at 8% (v/v) of the inoculum. This is already written clearly in Material and methods 2.3.
- I think the author should perform an experiment with a similar animal model to check the individual & conbination effect of Indicaxanthin, Chlorogenic Acid and 3-Hydroxymela-tonin on the osteoporosis.
Response: Thank you for the nice advice. You have a very good point, indeed further animal experiments are needed to check the individual & conbination effect of Indicaxanthin, Chlorogenic Acid and 3-Hydroxymelatonin in osteoporosis, and this is also the direction of our future experimental arrangements.
- I would like to add primary data from animal experiments, may in supplementary figures, such as water or food intake, body weight etc.
Response: Thanks. Primary data from animal experiments including food intake and body weight have been added in supplementary material.
Thank you and best regards.
Yours sincerely,
Xichun Peng, Ph.D
Jianming Luo, Ph.D
Email: [email protected]; [email protected]
